# Hepcidin Therapeutics

**DOI:** 10.3390/ph11040127

**Published:** 2018-11-21

**Authors:** Angeliki Katsarou, Kostas Pantopoulos

**Affiliations:** Lady Davis Institute for Medical Research, Jewish General Hospital, Department of Medicine, McGill University, Montreal, QC H3T 1E2, Canada; ageliki.katsarou@mail.mcgill.ca

**Keywords:** iron metabolism, hepcidin, ferroportin, hemochromatosis, anemia

## Abstract

Hepcidin is a key hormonal regulator of systemic iron homeostasis and its expression is induced by iron or inflammatory stimuli. Genetic defects in iron signaling to hepcidin lead to “hepcidinopathies” ranging from hereditary hemochromatosis to iron-refractory iron deficiency anemia, which are disorders caused by hepcidin deficiency or excess, respectively. Moreover, dysregulation of hepcidin is a pathogenic cofactor in iron-loading anemias with ineffective erythropoiesis and in anemia of inflammation. Experiments with preclinical animal models provided evidence that restoration of appropriate hepcidin levels can be used for the treatment of these conditions. This fueled the rapidly growing field of hepcidin therapeutics. Several hepcidin agonists and antagonists, as well as inducers and inhibitors of hepcidin expression have been identified to date. Some of them were further developed and are currently being evaluated in clinical trials. This review summarizes the state of the art.

## 1. Systemic Iron Homeostasis

Iron is an essential constituent of cells and organisms and participates in vital biochemical activities, such as DNA synthesis, oxygen transfer, and energy metabolism. The biological functions of iron are based on its capacity to interact with proteins and on its propensity to switch between the ferrous (Fe^2+^) and ferric (Fe^3+^) oxidation states. In spite of the high abundance of iron on the earth’s crust, its bioavailability is limited by the fact that under aerobic conditions, ferrous iron is readily oxidized to insoluble ferric. Accumulation of excess iron in cells is toxic, because “free” ferrous iron catalyzes the generation of hydroxyl radicals via Fenton chemistry, which attack and inactivate cellular macromolecules [1]. The intricate chemistry of iron poses a major challenge for iron homeostasis: To satisfy metabolic needs and limit toxic side effects [2]. Mammals have developed mechanisms to efficiently acquire and retain dietary iron in the body and store, but not excrete the iron excess.

Approximately 70% of body iron (~3–5 g in adult humans) is bound to heme and used in hemoglobin of red blood cells [3]. Another 2–3% is present in muscles, where it is mostly utilized in the heme moiety of myoglobin. Excess of body iron accumulates in liver hepatocytes, where it is stored within ferritin and can be mobilized for erythropoiesis under iron deficiency. On a daily basis, erythropoiesis requires up to 30 mg of iron, while non-erythroid cell requirements are ~5 mg. Bone marrow erythroblasts and non-erythroid cells in other tissues acquire iron from plasma transferrin. This protein not only serves as an iron carrier, but also keeps circulating iron in a redox-inactive state. Under physiological conditions, only ~30% of transferrin molecules are saturated with Fe^3+^ and the iron-free apo-transferrin offers a redox buffering capacity. The iron content of transferrin (~3 mg) represents a small, but a highly dynamic fraction of body iron, which turns over more than 10 times per day to satisfy metabolic needs. It is mainly replenished by iron recycled from effete red blood cells via tissue macrophages during erythrophagocytosis. Dietary iron absorption (1–2 mg/day) contributes to the buildup of iron stores during development; in adults it mainly serves to compensate for non-specific iron losses, for instance via bleeding or cell desquamation.

## 2. Hepcidin: The Key Iron Regulatory Hormone

Serum iron levels are under the control of hepcidin, a liver-derived peptide hormone and master regulator of iron metabolism [4]. It operates by binding to the iron exporter ferroportin in iron-releasing target cells, mainly tissue macrophages and duodenal enterocytes, but also other cell types (Figure 1). The binding of hepcidin occludes iron efflux [5] and triggers ubiquitination, internalization and lysosomal degradation of ferroportin [6]. This leads to intracellular iron retention and eventually to hypoferremia.

The *HAMP* gene encodes pre-pro-hepcidin, an 84 amino acids long precursor, which is primarily expressed by hepatocytes in the liver, and at much lower levels by other cells in extrahepatic tissues. Pre-pro-hepcidin is processed to pro-hepcidin upon removal of its endoplasmic reticulum targeting sequence, consisting of 24 N-terminal amino acids. Further cleavage at the C-terminus yields matures, bioactive hepcidin, an evolutionary conserved, cysteine-rich peptide of 25-amino acids with antimicrobial properties. It folds to a distorted β-sheet with an unusual disulfide bridge between adjacent C13-C14 at the turn of a hairpin loop; according to this model, the structure is stabilized by further disulfide bonding between C7-C23, C10-C22 and C11-C19 [7] (Figure 2, left). An alternative structural model postulates disulfide bond connectivity between C7-C23, C10-C13, C11-C19 and C14-C22 [8] (Figure 2, right). Interestingly, the structural organization of hepcidin based on disulfide bonding is not essential for iron-regulatory function, since the substitution of cysteines or the deletion of cysteine-containing segments do not impair hormonal activity [9,10].

Hepatocellular hepcidin expression responds to multiple stimuli, yet the major regulators are iron, inflammation and erythropoiesis [11]. Increases in serum or tissue iron trigger transcriptional induction of hepcidin via BMP/SMAD (Bone Morphogenetic Protein/Small Mothers Against Decapentaplegic) signaling (Figure 3). The mechanism involves secretion of *bone morphogenetic protein 6* (BMP6) from liver sinusoidal endothelial cells, which binds to type I (ALK2, ALK3, ALK6) and type II (ActRIIA, BMPRII) BMP receptors on hepatocytes and thereby activates the SMAD signaling cascade. Efficient iron signaling to hepcidin requires auxiliary factors, such as BMP2, the BMP co-receptor hemojuvelin (HJV), the hemochromatosis protein HFE, and the diferric transferrin sensor *transferrin receptor 2* (TfR2) [12]. The pathway is negatively controlled by the transmembrane serine protease matriptase-2 (also known as TMPRSS6), a hepcidin suppressor that appears to cleave HJV and other components of the hepcidin signaling pathway [13]. Iron-dependent upregulation of hepcidin serves to prevent excessive iron absorption when body iron stores are high.

Inflammatory cytokines promote hepcidin induction by several mechanisms. The most critical involves JAK/STAT (Janus Kinase/Signal Transducer and Activator of Transcription) signaling, which is activated in response to IL-6 (Figure 3). There is increasing evidence for a cross-talk between JAK/STAT and BMP6/SMAD signaling during inflammatory hepcidin induction [14,15,16]. Activin B, a JAK/STAT-induced ligand of BMP receptors can activate SMAD signaling to hepcidin [17], but its function is not essential [18]. On the other hand, BMP6 and HJV are critical components of the inflammatory hepcidin pathway [19,20], which is considered as an innate immune response to deprive invading bacteria of iron [21]. This is part of a broader iron withholding strategy within the context of “nutritional immunity” [22]. Under conditions of an increased erythropoietic drive, hepcidin expression is suppressed by erythroferrone (ERFE), a hormone secreted by erythroblasts in response to erythropoietin [23] that neutralizes BMP6 [24]. This allows iron mobilization for erythropoiesis; the involvement of further erythropoietin-induced hepcidin suppressors in this response is also possible. Dysregulation of hepcidin expression leads to “hepcidinopathies”, which are iron-related disorders of hepcidin deficiency or excess (Table 1).

## 3. Disorders with Hepcidin Deficiency

Hepcidin expression is suppressed in *hereditary hemochromatosis* (HH), in iron-loading anemias with ineffective erythropoiesis and in some chronic liver diseases.

### 3.1. Hereditary Hemochromatosis

Inactivating mutations in upstream regulators of hepcidin or direct disruption of hepcidin cause HH, an endocrine disorder of systemic iron overload [25]. As a result, hepcidin deficiency leads to increased dietary iron absorption (up to 8–10 mg/day) and uncontrolled release of iron to plasma, due to unrestricted expression of ferroportin in duodenal enterocytes and tissue macrophages. This promotes gradual saturation of plasma transferrin and buildup of unshielded *non-transferrin-bound iron* (NTBI), which accumulates within tissue parenchymal cells. Clinical complications of HH include liver cirrhosis, hepatocellular cancer, cardiomyopathy, diabetes mellitus, endocrinopathy, arthritis and osteoporosis. In addition, HH patients are susceptible to infection with siderophilic bacteria, but also various other pathogens [26,27].

HH is genetically heterogenous and its severity depends on the degree of hepcidin suppression relative to body iron stores [28]. The major form is linked to mutations in HFE (especially C282Y) and constitutes the most frequent genetic disorder in Caucasians. However, the clinical penetrance is variable and depends on further genetic and environmental factors. The disease phenotype is relatively mild, and symptoms typically develop after the fourth decade of life. Other forms of HH are rare and typically associated with more severe phenotypes. Thus, inactivation of either HJV or hepcidin cause *juvenile hemochromatosis* (JH) with early-onset iron overload in the late teens or early twenties. Inactivation of TfR2 yields an intermediate clinical phenotype compared to HFE-related HH and JH.

### 3.2. Iron-Loading Anemias

Hereditary or acquired anemias, such as thalassemias, congenital dyserythropoietic anemias, sideroblastic anemias or myelodysplastic syndromes, are associated with bone marrow hyperplasia and ineffective erythropoiesis [29]. Patients with severe forms of these diseases are often treated with blood transfusions, causing secondary iron overload. Ineffective erythropoiesis stimulates expression of ERFE and other factors, such as GDF15, which suppress hepcidin expression. This in turn promotes a hemochromatosis-like phenotype of iron overload in non-transfused patients with milder forms of disease; in addition, it aggravates the already existing secondary iron overload in transfused patients. Importantly, repressive erythropoietic signals dominate over the stimulating iron signals creating a vicious cycle in iron homeostasis and hepcidin expression [3].

### 3.3. Chronic Liver Diseases

Patients with *alcoholic liver disease* and *chronic hepatitis C* often exhibit some degree of hepatocellular iron overload that aggravates liver disease progression to advanced stages (cirrhosis and hepatocellular carcinoma). This is linked to hepcidin suppression by oxidative stress and possibly additional mechanisms. Hepatic iron overload is also observed in patients with *non-alcoholic fatty liver disease* (NAFLD). The pattern of excessive iron distribution is variable with preference in either hepatocytes or macrophages, which may underlie differential hepcidin regulation by positive and negative stimuli [30,31]. In advanced liver disease of any etiology, the expression of hepcidin decreases dramatically due to severe injury of hepatocytes, and hepcidin levels offer a potential biomarker for liver disease progression [32].

Interestingly, experimental data provided evidence for a protective role of hepcidin against liver fibrosis. Thus, adenoviral expression of hepcidin attenuated the development of liver fibrosis in mice subjected to CCl_4_ intoxication or to bile duct ligation [33]. The underlying mechanism involves a hepcidin-mediated increase in the iron content of hepatic stellate cells, which prevents their differentiation from collagen-secreting myofibroblasts. Another study using Tmprss6-/- mice showed that genetic hepcidin overexpression protected these animals against high fat diet-induced obesity and liver steatosis by stimulating lipolytic pathways [34]. These data highlight a critical role of hepcidin in liver disease pathogenesis.

## 4. Disorders with Hepcidin Resistance or Ferroportin Deficiency

*Ferroportin hemochromatosis* is a distinct clinical entity that develops as a result of “gain-of-function” ferroportin mutations that prevent the binding of hepcidin [35]. Conversely, “loss-of-function” ferroportin mutations that inhibit intracellular trafficking of the protein are the hallmark of *ferroportin disease*, which is characterized by macrophage iron loading. Hepcidin levels are elevated rather than suppressed in these ferroportin-associated disorders of hepcidin resistance, or ferroportin deficiency, respectively. It should be noted that both ferroportin hemochromatosis and ferroportin disease are transmitted in an autosomal dominant fashion, contrary to all other forms of HH, which are autosomal recessive.

## 5. Disorders with Systemic Hepcidin Overexpression

Systemic overexpression of hepcidin occurs in anemias with iron-restricted erythropoiesis, such as *iron-refractory iron deficiency anemia* (IRIDA), *anemia of inflammation* (AI; also known as anemia of chronic disease) or Castleman disease. Local overexpression of hepcidin has been reported in tumors.

### 5.1. Iron-Refractory Iron Deficiency Anemia

IRIDA is an autosomal recessive disease caused by loss-of-function mutations in the hepcidin suppressor matriptase-2 (TMPRSS6), which lead to hepcidin overexpression [36]. It is characterized by hypochromic microcytic anemia, hyperhepcidinemia, hypoferremia, low transferrin saturation and unresponsiveness to oral iron therapy.

### 5.2. Anemia of Inflammation

AI is a heterogenous disorder caused by chronic inflammation due to infections, inflammatory bowel disease, inflammatory rheumatic disease, *chronic kidney disease* (CKD) or obesity [37]. It is also observed in cancer patients (see Section 6) and in frail, elderly persons. AI is the most common type of anemia among hospitalized patients in the developed world. Unresolved inflammatory induction of hepcidin is one of the contributors to AI, because it causes hypoferremia and iron sequestration in tissues, limiting its availability for erythropoiesis. Other contributing factors are reduced proliferation of erythroblasts, reduced expression of erythropoietin, impaired erythropoietin signaling and increased erythrophagocytosis. In CKD, hyperhepcidinemia is aggravated by ineffective renal clearance [38]. AI is normochromic and normocytic, but this phenotype can be affected by chronic blood loss or dietary iron malabsorption, which result in smaller and hypochromic red blood cells [39].

### 5.3. Castleman Disease

Castleman disease is a rare lymphoproliferative disorder characterized by generalized lymphadenopathy and multiple organ involvement that is linked to chronic overproduction of IL-6 [40]. Castleman disease patients present with iron-refractory hypochromic microcytic anemia, which develops in response to IL-6-mediated upregulation of hepcidin [41].

## 6. Disorders with Local Hepcidin Overexpression

Hepcidin is produced at lower levels in several extrahepatic tissues, including the heart or the brain, where it exerts local cell autonomous functions [42,43]. Interestingly, many tumor cells produce and utilize hepcidin for their own advantage. Thus, hepcidin targets ferroportin in tumor cells in an autocrine manner and thereby promotes retention of iron, which is essential for cell proliferation and tumor growth [44]. This mechanism has been documented in breast [45], prostate [46] and thyroid [47] cancers. It should also be noted that in advanced disease stages, the immune response against primary or metastatic tumor cells may trigger hepcidin induction in hepatocytes and thereby lead to systemic hepcidin overproduction. This may result in AI due to hepcidin-mediated iron sequestration in tissues and inhibition in iron absorption.

## 7. The Need for Hepcidin Therapeutics

Targeting the hepcidin-ferroportin axis could offer therapeutic benefits to patients with disorders related to aberrant hepcidin expression [48,49,50]. Hepcidin therapeutics could complement or even replace current treatment modalities, which have several limitations.

### 7.1. Narrowing the Management Gap in Iron Overload Disorders Linked to Hepcidin Deficiency

The standard of care for HH is therapeutic phlebotomy, which reduces circulating iron burden and also promotes iron mobilization from stores [51]. This approach is effective and essentially normalizes the life-span of HH patients without liver cirrhosis and diabetes [52]. However, there is no evidence base for optimal start time, frequency, duration and endpoint of phlebotomy. Moreover, this treatment cannot reverse liver cirrhosis, diabetes, arthritis, cardiomyopathy or hypogonadism, while a few patients may not tolerate, or exhibit low compliance, to life-long treatment. Another limitation is that phlebotomy stimulates dietary iron absorption by further suppressing hepcidin via erythropoietic regulators.

Iron overload in iron-loading anemias is managed by iron chelation therapy. Clinically approved iron chelating drugs include desferrioxamine (DFO), deferiprone (DFP) and deferasirox (DFX). DFO was first utilized for the treatment of transfused β-thalassemia patients in the 1970s. Due to the accumulated >40-year long experience and its low cost, it remains a first line therapeutic option in many guidelines [53]. However, DFO is poorly absorbed in the gastrointestinal tract and has a short (~20 min) plasma half-life. Thus, to reach effective pharmacological concentrations the drug is administered parenterally with the aid of a portable infusion pump at least 4–5 days per week for 8-10 h each time. The cumbersome procedure reduces compliance and significantly compromises the quality of patients’ life. This prompted the development of the orally absorbed iron chelators DFP and DFX, which exhibit better pharmacokinetics, with plasma half-lives of 1–3 h and 8–16 h, respectively. Nevertheless, the use of iron chelating drugs is associated with the risk of side effects, such as neutropenia/agranulocytosis, skin rash, gastrointestinal disturbances, retinopathy or bone abnormalities. Another problem is that these drugs cannot prevent dietary iron absorption, which is highly induced in patients with iron-loading anemias, due to erythropoietic suppression of hepcidin.

Thus, hepcidin replacement therapy could provide an etiologic cure of HH, and correct aggravating dietary iron absorption in transfusional iron overload. Proof of principle is provided by experiments with mouse models of HH (Hfe-/-) and β-thalassemia (Hbb^th3/+^). Thus, iron overload was corrected when these animals were crossed with mice overexpressing transgenic hepcidin [54,55], or with Tmprss6-/- mice overexpressing endogenous hepcidin [56,57]. Moreover, the increased hepcidin expression by these manipulations led to improved erythropoiesis in Hbb^th3/+^ mice [55,57].

In light of the experimental evidence for the protective effects of hepcidin against liver fibrosis and obesity [33,34], hepcidin agonists, or inducers of hepcidin expression, could also be used for the treatment of these highly prevalent disorders. Finally, hepcidin agonists/inducers could enhance “nutritional immunity” during infection with extracellular pathogens.

### 7.2. Narrowing the Management Gap in Anemias and Other Disorders Linked to Hepcidin Overexpression

IRIDA patients are typically managed with intermittent parenteral iron therapy that only partially corrects anemia [36]. Improvement may need repeated dosing, which is associated with potential side effects and the risk for reticuloendothelial iron overload. Control of hepcidin overexpression by using hepcidin antagonists is expected to improve responses of IRIDA patients not only to parenteral, but also oral iron formulations.

AI can be corrected by treating the underlying cause of the primary disease, but this is not always feasible. Alternatively, AI patients can be treated with oral or parenteral iron administration, which increases the risk of bacteremia and allergic reactions [58]. Another option is the administration of *erythropoiesis stimulating agents* (ESA) based on recombinant human erythropoietin, often together with intravenous iron; however, prolonged use of ESAs may lead to cerebrocardiovascular complications [59]. When AI is combined with severe iron deficiency, blood transfusion is the preferred treatment; however, this is limited by the risk of infections, immune reactions or lung injury [60]. A major drawback that compromises efficiency of all these approaches is the inflammatory overexpression of hepcidin, which sequesters iron to tissues and reduces its availability for erythropoiesis. Thus, iron/ESA therapy could be complemented (and in some cases even replaced) by the use of hepcidin antagonists.

Proof of principle has been provided by pharmacological experiments showing improvement of anemia following hepcidin neutralization in a rat model of AI, based on injection with *group A streptococcal peptidoglycan-polysaccharide* (PG-APS) [15,61]. Moreover, inoculation of Hamp-/- or IL6-/- mice with heat-killed *Brucella abortus* yielded a milder AI phenotype [62,63].

## 8. Inducers of Hepcidin Expression

Various molecules exhibit pharmacological potential to induce hepcidin expression. These include recombinant BMP6, TMPRSS6-silencing oligonucleotides, and a wide range of natural or synthetic small molecules. A detailed list is shown in Table 2. Clinically relevant hepcidin inducers currently undergoing evaluation in randomized controlled trials for disorders with hepcidin deficiency are highlighted in Figure 1.

### 8.1. Recombinant BMP6

Treatment of Hfe-/- mice with the major hepcidin inducer BMP6 was shown to increase hepatic hepcidin expression, which in turn caused a decrease in serum iron levels, due to iron retention in the spleen and the duodenum [64]. However, this was associated with peritoneal calcifications, possibly because BMP6 was administered intraperitoneally. Based on these findings, the off-target effects of BMP6 and other BMPs in bone metabolism appear to limit their clinical applications in the context of iron overload.

### 8.2. TMPRSS6-Silencing Oligonucleotides

Silencing of Tmprss6 mRNA using lipid nanoparticle formulated small interfering RNAs (siRNAs) or *antisense oligonucleotides* (ASOs) was shown to induce hepcidin and improve the phenotypes of hemochromatotic Hfe-/- and thalassemic Hbb^th3/+^ mice [65,66]. Of note, combined treatments of Hbb^th3/+^ mice with Tmprss6 siRNAs or ASOs, together with the oral iron chelator deferiprone, resulted in better correction of erythropoiesis and prevention of secondary iron overload [67,68]. When conjugated with *triantennary N-acetyl galactosamine* (GalNAc) for targeted delivery to hepatocytes, the Tmprss6-ASOs exhibited improved efficacy [69]. GalNAc-conjugated Tmprss6 siRNAs demonstrated a therapeutic potential in splenectomised Hbb^th3/+^ mice [70], a finding of clinical interest for splenectomised thalassemic patients. Ionis Pharmaceuticals Inc. developed TMPRSS6-silencing molecules with the commercial name IONIS-TMPRSS6-L_RX_, which are currently being tested in phase 1 clinical trials (ClinicalTrials.gov Identifier: NCT03165864).

### 8.3. Small Molecule Hepcidin Inducers

Several small molecules were shown to activate hepcidin pathways. A high-throughput chemical screen in zebrafish identified three steroid molecules i.e., progesterone, epitiostanol and mifepristone as potent hepcidin inducers. They operate via a pathway involving the *progesterone receptor membrane component-1* (PRMC-1) [71]. A small-scale chemical screen in zebrafish embryos uncovered the hepcidin-inducing capacity of genistein, a phytoestrogen that is abundant in soybean. Genistein was shown to upregulate hepcidin in HepG2 hepatoma cells and zebrafish embryos by promoting STAT3 phosphorylation [72].

A subsequent high-throughput screen of small molecules in HepG2 cells identified ipriflavone and vorinostat, which are synthetic drugs that inhibit bone resorption or histone deacetylase activity and are clinically applied for the treatment of osteoporosis or cutaneous T cell lymphoma, respectively. They both stimulate expression of hepcidin and other BMP- and STAT3-dependent genes without affecting SMAD1/5/8 or STAT3 phosphorylation, and exhibit 10-fold higher potency than genistein [73]. Ipriflavone was further tested in vivo and was shown to alleviate dietary iron overload in wild type mice; however, it failed to reduce iron overload in Hbb^th3/+^ mice [74]. The hepcidin-stimulating effects of vorinostat in Huh7 hepatoma cells were also validated in another study [75], and are consistent with earlier data showing that the histone deacetylase inhibitor trichostatin A induces hepcidin by inhibiting the binding of C/EBPα and STAT3 in the *HAMP* promoter [76]. Nevertheless, vorinostat failed to induce hepcidin in mice [75]. The non-steroidal anti-inflammatory drug diclofenac likewise stimulated hepcidin expression in vitro, but not in vivo [75]. In a screen of natural compounds, icariin, a plant flavonol glycoside, was documented to stimulate hepcidin expression in HepG2 cells by inducing SMAD1/5/8 or STAT3 phosphorylation. These findings were validated in mice, while similar results were obtained with the icariin analogue epimedin C [77]. Several polyphenolic small molecules or phytoestrogens that are found in fruits and vegetables likewise induced hepcidin in HepG2 cells and in rats. These include resveratrol, quercetin, kaemferol, naringenin, epi-gallo-catechin-3-gallate, and operate by activating Nrf2 for binding to an *antioxidant response element* (ARE) in the *HAMP* promoter [78].

A genetic siRNA screen revealed that Ras/Raf/MAPK and PI3K/Akt/mTOR signaling suppress hepcidin; consequently, pharmacological inhibitors of these pathways, such as sorafenib, wortmannin, rapamycin and metformin were shown to induce hepcidin in hepatoma cells and primary hepatocytes [79]. In another setting, dietary supplementation of Hfe-/- mice with adenine was reported to induce hepcidin and attenuate iron overload by a mechanism requiring BMP/SMAD and *cAMP/protein kinase A* (PKA) signaling [80]. While some data obtained with small molecule hepcidin inducers are interesting, the potential for these drugs for clinical application in the context of hepcidin deficiency appears limited, due to lack of specificity.

## 9. Hepcidin Agonists

Administration of synthetic hepcidin can effectively reduce plasma iron levels in wild type [81] or Hfe-/- mice [82]. However, the clinical value of this approach is limited by the high cost of manufacturing correctly folded, bioactive hepcidin, and by the rapid clearance of native hepcidin peptide in the blood. This prompted the development of cheaper hepcidin derivatives with improved pharmacological and pharmacokinetic properties. A detailed list is shown in Table 3a. Clinically relevant hepcidin agonists currently undergoing evaluation in randomized controlled trials for disorders with hepcidin deficiency are highlighted in Figure 1.

### 9.1. Minihepcidins

Mutational studies revealed that the N-terminus of hepcidin is necessary and sufficient for binding to ferroportin, and the remaining sequence is permissive to mutations [9]. This paved the way for the development of stable, bioactive truncated hepcidin derivatives. The first generation minihepcidins consist of 7 to 9 N-terminal amino acids with a free sulfhydryl group at C7; among them Hep9 was the most potent in in vitro ferroportin internalization assays [83]. Minihepcidins were further optimized by chemical modifications. To improve stability, peptides were circularized, but this resulted in decreased potency compared to Hep9. For protection against proteolysis, unusual amino acids, including N-substituted, beta-homo, and D-amino acids were introduced to the initial chemical structure resulting in retro-inverso (ri-) analogues (i.e., reversed sequence of parental L-peptides). To improve intestinal absorption, ri- analogues were conjugated to fatty acids (palmitoyl- groups) or chenodeoxycholic or ursodeoxycholic bile acids (cheno- and urso- groups, respectively). Modified minihepcidins where then tested in vivo. Injection of unmodified Hep9 into wild type mice failed to cause any effect on serum iron. By contrast, ri-hep9 and palmitoyl-ri-hep9 promoted hypoferremia, while high doses of the latter had almost the same effect on serum iron levels to equivalent doses of full-length hepcidin [83].

Administration of PR65, another optimized minihepcidin, prevented iron loading and corrected iron distribution in previously iron-depleted Hamp-/- mice. However, excessive amounts of the compound led to iron-restrictive anemia, underlying the need for dose optimization. The effects of PR65 in iron-loaded Hamp-/- mice were less compelling, and limited to partial iron re-distribution from the liver to the spleen [84].

Because free sulfhydryl groups exhibit non-specific reactivity and may cause dermatological side effects, an S-protection strategy was used to develop the PR73 minihepcidin [85]. Daily administration of PR73 to Hamp-/- mice for one week led to significant iron redistribution from pancreatic acinar cells to macrophages [86]. Attempts to further improve PR73 were made. Cyclization resulted in agonist mHS17, the most active analogue of PR73, but this was several times less active than PR73 in vitro and ineffective in vivo [87]. Thiol-protection with activated vinyl thioethers yielded the S-vinyl-PR73SH derivative, which exhibited almost the same potency with the parental compound in vitro and in vivo [85].

PR73 was also evaluated for possible protective effects in mouse models of bacterial infections. Treatment of Hamp-/- mice with PR73, pre- or 3 h post-infection with siderophilic *Vibrio vulnificus*, caused hypoferremia and eventually repressed bacterial growth and reduced mortality [88]. Similar results were obtained in Hamp-/- or iron-loaded wild type mice infected with siderophilic *Yersinia enterocolitica* [89], or non-siderophilic *Klebsiella pneumoniae* or *Escherichia coli* [90,91]. These data uncover a therapeutic potential of minihepcidins, and probably other hepcidin agonists, against bacterial infections.

Merganser Biotech Inc., a start-up company, developed minihepcidins M004 and M009. The latter is a prodrug with an extra *S*-methyl group at C7, which is converted to M004 upon S-methyl removal in the blood. Daily injections of young Hbb^th3/+^ mice with M004, or biweekly injections of old Hbb^th3/+^ mice with M009, improved erythropoiesis and reduced splenomegaly and iron accumulation in the liver and kidneys within 6 weeks [92]. Further improvement was observed when M009 administration was combined with the iron chelator deferiprone. Interestingly, M009 was also shown to reduce erythrocytosis and splenomegaly in a mouse model of polycythemia vera [92].

### 9.2. Other Hepcidin Derivatives

La Jolla Pharmaceutical Company developed LJPC-401, a proprietary formulation of hepcidin. The molecule completed phase 1 clinical trials and has initiated phase 2 in patients with HH (ClinicalTrials.gov Identifier: NCT03395704) or transfusion-dependent β-thalassemia with myocardial iron overload (ClinicalTrials.gov Identifier: NCT03381833). Data from phase 1 show that LJPC-401 effectively reduces serum iron levels in a dose-dependent manner with a maximum at 8 h post injection, and in the absence of significant adverse effects [93].

Protagonist Therapeutics Inc. launched PTG-300, another proprietary hepcidin formulation. The compound reduces serum iron in cynomolgus monkeys and improves erythropoiesis in the Hbb^th3/+^ mouse model of β-thalassemia [94]. PTG-300 recently completed phase 1 clinical trial in healthy volunteers. The data show that the drug is well tolerated and reduces serum iron in a dose-dependent manner; notably this response is sustained for 6 days [95]. A phase 2 clinical trial evaluating the effects of PTG-300 in β-thalassemia patients is expected to start in the fourth quarter of 2018 (http://www.protagonist-inc.com/randd-pipeline.php#ptg300).

## 10. Inhibitors of Ferroportin Activity

Theoretically, the pharmacological effects of hepcidin replacement therapy could also be achieved by inhibiting the synthesis or the iron-exporting activity of ferroportin. Moreover, these approaches would be appropriate for the treatment of ferroportin hemochromatosis, which is caused by hepcidin resistance rather than deficiency. Currently, there is no reported attempt for therapeutic targeting of ferroportin synthesis, but efforts have been made to develop ferroportin inhibitors. Thus, Vifor Pharma has generated an orally administered small molecule that binds to ferroportin and inhibits iron efflux (Table 3b and Figure 1). The drug (VIT-2763) was expected to enter phase 1 clinical trial in early 2018 (http://www.viforpharma.com/en/media/press-releases/201802/2167159).

## 11. Inhibitors of Hepcidin Expression

Several inhibitors of hepcidin expression have been identified. These include macromolecular inhibitors of BMP6 or HJV, small molecule inhibitors of BMP/SMAD signaling, neutralizing antibodies against IL-6 receptor or IL6, small molecule inhibitors of JAK/STAT3 signaling, sex hormones, and vitamin D. A detailed list is shown in Table 4. Clinically relevant hepcidin inhibitors currently undergoing evaluation in randomized controlled trials for AI are highlighted in Figure 1.

### 11.1. Inhibitors of BMP6/HJV

BMP6 and HJV are essential for appropriate hepcidin induction during inflammation [19,20]. Thus, targeting these proteins is expected to be beneficial in disorders associated with hepcidin overexpression. Eli Lilly and Company developed LY3113593, a monoclonal humanized antibody against BMP6, for the treatment of anemia in patients with CKD. The BMP6-neutralizing drug has been evaluated for safety and pharmacokinetics in two phase 1 clinical trials with healthy volunteers and CKD patients (ClinicalTrials.gov Identifiers: NCT02144285 and NCT02604160). The studies are completed, but the results are not yet publicly available.

Another BMP6-blocking strategy involves heparin, a glycosaminoglycan produced by mast cells, which is used in clinical settings as an anticoagulant. Biochemical experiments showed that heparin binds with high affinity to BMP6 and abolishes its hepcidin signaling function. Moreover, administration of heparin to mice downregulated hepcidin expression and increased serum iron, while a similar phenotype was documented in patients treated with heparin to prevent deep vein thrombosis [96]. Heparin was also shown to inhibit hepcidin expression in *Mycobacterium tuberculosis*-infected human macrophages [97]; however, the role of macrophage hepcidin in AI is not well understood. Modified glycol-split and oversulfated heparins lacking coagulant activity retain hepcidin suppressive properties. Thus, treatment with these drugs improved anemia in a heat-killed *Brucella abortus* mouse model of AI, and even further suppressed hepcidin in Bmp6-/- mice [98,99]. Clearly, these promising results highlight the need for comprehensive evaluation of heparin derivatives for efficacy and safety in clinical trials.

Erythroferrone was recently identified as a potent competitive inhibitor of BMP6 [24]. This finding raises the interesting possibility for using this hormone to treat anemias with excessive hepcidin expression. The pharmacological potential of erythroferrone awaits experimental validation. Nevertheless, since erythroferrone moonlights as myokine (myonectin/CTRP15) with metabolism-modulating properties [23,100], potential pleiotropic effects should be considered.

A soluble HJV construct fused with the Fc domain of IgG (sHJV.Fc) specifically inhibits HJV-mediated induction of hepcidin in cultured hepatoma cells and in mice by competitive binding to BMP6 [101,102]. Moreover, sHJV.Fc was shown to correct hepcidin levels and anemia in the PG-APS rat model of AI [15]. FerruMax Pharmaceuticals, a start-up biotech company, aimed to investigate the pharmacological potential of sHJV.Fc for the treatment of AI in two phase 1 clinical trials with CKD patients (ClinicalTrials.gov Identifiers: NCT01873534 and NCT02228655). However, the trials were terminated early due to “inability to recruit patients meeting eligibility criteria”.

Abbvie developed humanized monoclonal antibodies against HJV. A single-dose of the HJV-neutralizing antibodies ABT-207 or H5F9-AM8 in rats and cynomolgus monkeys had long-lasting suppressing effects on hepcidin and promoted an increase in serum iron [103]. H5F9-AM8 was further evaluated in preclinical models of AI (chronic arthritis in rats, aseptic inflammation in mice) and IRIDA (Tmprss6-/- mice). In all conditions tested, the antibody suppressed hepcidin and improved anemia [104]. H5F9-AM8 also reduced systemic and local tumor hepcidin in a mouse xenograft model; however, the suppression of hepcidin expression in cancer cells was not associated with inhibition of tumor growth [105].

### 11.2. Small Molecule Inhibitors of BMP/SMAD Signaling

Dorsomorphin was identified in a chemical screen as the first small molecule inhibitor of the BMP/SMAD signaling pathway [106], which is essential for inflammatory induction of hepcidin [14,15,16]. It operates by targeting type I BMP receptors, thereby blocking phosphorylation of SMAD1/5/8. The optimized dorsomorphin derivative LDN-193189 inhibited BMP/SMAD signaling to hepcidin in hepatocytes, which in turn stimulated erythropoiesis and attenuated anemia in the PG-APS rat model of AI [15,61]. Orally administered LDN-193189 was also shown to increase hemoglobinization in turpentine-challenged mice [107]. Surprisingly, hepcidin induction by activin B, which is mediated by ALK2 and the type II BMP receptor ActRIIA, is insensitive to LDN-193189 [108]. A major limitation of LDN-193189 as hepcidin inhibitor is its broad specificity that leads to off-target effects [109,110]. LDN-212854, another dorsomorphin derivative with increased selectivity to ALK2 versus ALK3, was shown to suppress IL-6-induced hepcidin expression in HepG2 cells, yet less effectively compared to LDN-193189 [111].

Further chemical screens identified additional compounds with inhibitory activity on BMP/SMAD signaling to hepcidin. The dietary flavonoid myricetin was shown to prevent lipopolysaccharide-induced hypoferremia in mice [112]. An indazole molecule was derivatized to several indazole-based inhibitors [113], including DS28120313 [114] and orally active DS79182026, which could antagonize induction of hepcidin in mice injected with IL-6 [115]. Phenotypic screening by applying chemical proteomics and a radioactive compound-binding assay in HepG2 cells identified ALK2 and ALK3 as the primary targets of DS79182026 [116].

Tolero Pharmaceuticals, Inc., Lehi, UT, USA, launched TP-0184, a small molecule inhibitor of ALK2. The drug is currently undergoing a phase 1 clinical trial to determine its maximum tolerated dose and dose-limiting toxicities in patients with advanced solid tumors (ClinicalTrials.gov Identifier: NCT03429218). In preclinical mouse models of inflammation- and cancer-induced anemia, TP-0184 decreased hepatic hepcidin mRNA and improved hemoglobinization [117,118].

The value of ALK2 as a critical regulator of inflammatory hepcidin induction has been verified by independent studies using momelotinib, a JAK1/2 inhibitor that is under clinical evaluation for the treatment of myelofibrosis. A completed phase 1/2 clinical trial with myelofibrosis patients showed that this drug also improves anemia [119]. This finding was surprising because JAK1/2 signaling is required for erythropoiesis stimulation via erythropoieitin [120]. Experiments with the PG-APS rat model of AI confirmed that momelotinib ameliorates anemia in response to hepcidin inhibition; moreover, they demonstrated that this response is caused by momelotinib-mediated targeting of ALK2 [121].

Evaluation of data from a genome-wide RNAi screen for hepcidin regulators, with a focus on targets of clinically applied drugs, identified spironolactone and imatinib as hepcidin suppressors [75]. Spironolactone is an aldosterone antagonist commonly used to treat hypertension and congestive heart failure, while imatinib is a tyrosine kinase inhibitor that is used in cancer therapy. Both drugs appear to reduce hepcidin expression in hepatoma cells, in primary hepatocytes and in mice. The mechanism is not well understood, but appears to require intact BMP/SMAD signaling.

### 11.3. Neutralizing Antibodies against IL-6 Receptor or IL-6

IL-6 is critical for inflammatory induction of hepcidin, and Il6-/- (IL-6 knockout) mice exhibit an impaired hypoferremic response to acute inflammation [122]. Thus, neutralizing IL-6 or its receptor offers another potential option for the management of AI. Tocilizumab, a humanized anti-IL-6 receptor antibody, has a good safety profile and is used as an immunosuppressive drug mainly for the treatment of rheumatoid arthritis [123]. Clinical studies showed that it can also significantly reduce serum hepcidin and improve anemia in rheumatoid arthritis patients [124,125,126]. In addition, tocilizumab [127], as well as the anti-IL-6 antibody siltuximab [128], lowered hepcidin and ameliorated anemia in multicentric Castleman disease patients. Tocilizumab or another IL-6 receptor antibody (MR16-1) had similar therapeutic effects in a cynomolgus monkey model of collagen-induced arthritis [129], and in a mouse model of cancer-related anemia [130], respectively. Nevertheless, the high cost of these drugs may restrict their applicability beyond the treatment of rheumatoid arthritis and Castleman disease.

### 11.4. Small Molecule Inhibitors of JAK/STAT3 Signaling

There are no appropriate small molecule inhibitors of hepcidin that specifically target the JAK/STAT3 signaling pathway downstream of IL-6 and the IL-6 receptor. Another caveat is that any specific inhibitors of JAK1/2 would also be incompatible with the requirement of this protein in erythropoietin signaling [120]. STAT3 inhibitors, such as curcumin, AG490 or PpYLKTK were reported to inhibit inflammatory induction of hepcidin in cultured cells [131], and in case of AG490 also in mice [132]. Moreover, a single oral dose of curcuma containing 2% curcumin decreased hepcidin in healthy volunteers; however, this had no effect on serum iron [133]. Acetylsalicylic acid (aspirin) was shown to downregulate hepcidin by decreasing JAK2 and STAT3 phosphorylation in murine BV-2 microglia [134] and rat PC12 pheochromocytoma cells [135], but the relevance of this finding to hepatocellular hepcidin is unknown. Generally, STAT3 inhibitors exhibit low target specificity and selectivity, and therefore do not appear optimal candidates for the treatment of AI. Nevertheless, some promising data have been obtained in preclinical models.

For instance, maresin 1, a macrophage-derived ω-3 fatty acid metabolite, exhibited protective anti-inflammatory effects in Il10-/- (IL-10 knockout) mice, which develop colitis with iron deficiency anemia [136]. Further experiments showed that maresin 1 inhibits hepcidin by reducing STAT3 phosphorylation, and thereby ameliorates anemia in this model [137]. A polysaccharide isolated from the roots of the medicinal herb *Angelica sinensis* was found to repress hepcidin expression in HepG2 cells and to relieve anemia in rat models of AI. The polysaccharide appears to operate by inhibiting both JAK/STAT3 and BMP/SMAD signaling pathways [138,139,140].

Hydrogen sulfide (H_2_S) is a gasotransmitter with signaling properties [141]. Pharmacological administration of H_2_S was shown to suppress hepcidin in mouse models of acute lipopolysaccharide-induced or chronic turpentine-induced inflammation; mechanistic experiments revealed that H_2_S inhibits IL-6 production and STAT3 phosphorylation in response to IL-6 signaling [142,143]. Further studies showed that H_2_S promotes JAK2 degradation by a pathway involving adenosine *5’-monophosphate-activated protein kinase* (AMPK), a metabolic enzyme that controls intracellular AMP/ATP ratio. Moreover, activation of AMPK by metformin, a widely used antidiabetic drug, inhibited hepcidin expression and thereby relieved hypoferremia and anemia in mouse models of inflammation. Notably, metformin intake was also associated with low circulating hepcidin and reduced anemia morbidity in diabetic patients [144]. On the other hand, activation of AMPK by metformin was reported to induce hepcidin in cultured Huh7 hepatoma cells [79]; these seemingly conflicting data highlight the need for more mechanistic studies.

An in-silico screen identified *guanosine 5′-diphosphate* (GDP) as a hepcidin-binding molecule. GDP was further shown to operate as a competitive inhibitor of hepcidin that prevents hepcidin-mediated internalization of ferroportin in cultured cells. GDP (combined with ferrous sulfate) also inhibited hepcidin and improved anemia in a mouse model of inflammation, which was partly due to reduced STAT3 phosphorylation [145].

### 11.5. Sex Hormones

Testosterone, the primary male sex hormone and anabolic steroid, inhibits hepcidin by enhancing *epidermal growth factor receptor* (EGFR) signaling [146] and by reducing BMP/SMAD signaling [147]. Along these lines, testosterone therapy is known to cause erythrocytosis [148]. In female heat-killed *Brucella abortus*-inoculated mice, administration of testosterone improved hemoglobinization by suppressing hepcidin and stimulating erythropoiesis [149]. Similar effects were observed in clinical trials with healthy or anemic men [150,151,152] and men with type 2 diabetes and concurrent hypogonadotropic hypogonadism [153]. Conversely, androgen deprivation therapy reduced erythropoiesis and increased hepcidin in men with prostate cancer [154]. These findings suggest that testosterone could be valuable for the treatment of AI; however, cardiovascular adverse effects associated with testosterone administration limit its clinical applications [155].

17β-Estradiol, the major female sex hormone, was shown to suppress hepcidin transcription in Huh7 and HepG2 hepatoma cells via an estrogen responsive element half-site located in *HAMP* promoter. Nonetheless, while administration of 17β-estradiol to mice reduced hepcidin, it did not affect serum iron levels [156]. These findings are in line with a study demonstrating a significant reduction in serum hepcidin without correlation with serum iron in female patients with elevated estrogens, due to fertility treatments [157]. On the other hand, in a study with premenopausal women during various phases of their monthly cycle, serum 17β-estradiol was negatively correlated with hepcidin and positively correlated with iron [158]. Definitely, more work is required to clarify the mechanism and physiological relevance of hepcidin regulation by estrogens before considering any therapeutic applications.

### 11.6. Vitamin D

Vitamin D exerts various biological functions upon binding to *vitamin D receptor* (VDR); these include the stimulation of intestinal absorption of calcium, magnesium and phosphate, but also modulation of signaling pathways [159]. There is increasing evidence that vitamin D levels are inversely associated with anemia in chronic inflammatory diseases [160]. Experiments in cultured hepatoma cells, monocytic cells and peripheral blood monocytes demonstrated that vitamin D inhibits expression of inflammatory cytokines; and suppresses hepcidin by a mechanism involving VDR binding to the *HAMP* promoter [161,162]. Moreover, vitamin D supplementation to healthy volunteers decreased circulating hepcidin levels [161]. This finding was further validated in randomized controlled trials with healthy adults [163], stage 2–3 CKD patients [162] and mechanically ventilated critically ill patients [164]. However, in other randomized controlled trials, vitamin D supplementation failed to correct hepcidin in non-dialysis stage 2–5 pediatric CKD patients [165] and stage 3–4 adult CKD patients [166]. These controversial findings do not provide compelling evidence that vitamin D has a pharmacological value as a hepcidin-lowering agent in AI. Nevertheless, they may be confounded by differences in nutritional forms of vitamin D, dosage, and severity of disease. The interest in this area of research remains high, as illustrated by several clinical trials that are currently underway (ClinicalTrials.gov Identifiers: NCT03001687, NCT03145896, NCT02714361, NCT03166280, NCT01632761, NCT03718182, NCT03472833), which may provide more conclusive answers.

## 12. Hepcidin Antagonists

Hepcidin can be antagonized directly by hepcidin-binding inhibitors, which neutralize hepcidin and reduce its availability to interact with ferroportin. Alternatively, hepcidin can be antagonized indirectly by molecules that bind to ferroportin and competitively inhibit the binding of hepcidin without affecting iron export. A detailed list is shown in Table 5. Clinically relevant hepcidin antagonists currently undergoing evaluation in randomized controlled trials for AI are highlighted in Figure 1.

### 12.1. Direct Hepcidin Inhibitors

Administration of human hepcidin antibodies effectively neutralized hepcidin and triggered responses to hepcidin deficiency in human hepcidin knock-in mice [167,168] and in cynomolgus monkeys [168,169]. Furthermore, the therapeutic use of hepcidin antibodies improved hemoglobinization in the heat-killed *Brucella abortus* model of AI, established in hepcidin knock-in mice [167,168]. Eli Lily and Company developed LY2787106, a human hepcidin monoclonal antibody. Its safety profile and efficacy to improve erythropoiesis were evaluated in a phase 1 randomized controlled clinical trial with patients having cancer-associated anemia. LY2787106 was well tolerated and promoted an increase in serum iron and transferrin saturation within 24-48 h. However, the changes were temporary, and values returned to baseline within a week after dosing. Moreover, there was no correction of anemia, even when the LY2787106 treatment was combined with iron supplementation [170].

Anticalins are engineered polypeptides that derive from human lipocalins and mimic antibody activity by binding to protein targets [171]. Like other therapeutic proteins, anticalins can be PEGylated (coupled with polyethylene glycol) to improve pharmacological properties, such as solubility, stability and plasma half-life [172]. Pieris Pharmaceuticals Inc launched PRS-080, a hepcidin-neutralizing PEGylated anticalin. The safety profile and efficacy of PRS-080 were verified in a phase 1 clinical trial with healthy subjects. Apart from mild adverse effects (mostly headache) in some patients, PRS-080 was well tolerated and decreased hepcidin one hour after infusion. This in turn led to increased serum iron, transferrin saturation and ferritin [173]. PRS-080 is currently being evaluated in phase 1b and 2a clinical trials with CDK patients undergoing hemodialysis (ClinicalTrials.gov Identifiers: NCT02754167 and NCT03325621, respectively).

Spiegelmers are synthetic aptamers made of L-oligoribonucleotides, which are engineered to bind with high affinity and selectivity to low-molecular weight protein or peptide targets [174]. NOXXON Pharma AG developed NOX-H94 (or Lexaptepid Pegol), a PEGylated Spiegelmer that binds to and inactivates hepcidin. In preliminary tests NOX-H94 effectively antagonized hepcidin-induced ferroportin internalization in cultured J774 macrophages, and reversed inflammation-related hypoferremia in cynomolgus monkeys [175]. In a phase 1 randomized control clinical trial with healthy volunteers, NOX-H94 inhibited hepcidin in a dose-dependent manner and promoted increases in serum iron levels with only minor adverse effects (mild increase in transaminases at high doses) [176]. In another phase 1 trial with human volunteers who developed endotoxemia following lipopolysaccharide injections, NOX-H94 neutralized high levels of hepcidin and corrected endotoxin-induced hypoferremia [177]. In a phase 2b clinical trial with ESA-hyporesponsive CKD patients undergoing hemodialysis, NOX-H94 significantly elevated serum iron and improved hemoglobinization [178]. Similar effects were observed in a phase 2a clinical trial with patients having cancer-related anemia [179].

### 12.2. Ferroportin-Binding Hepcidin Inhibitors

High-throughput screens have been performed to identify small molecule ferroportin-binding hepcidin inhibitors. In one of them the best hit was fursultiamine, a thiamine (vitamin B_1_) derivative. This drug irreversibly reacts with the C326 thiol residue in ferroportin that is essential for binding to hepcidin. Consequently, fursultiamine efficiently prevented hepcidin-mediated internalization of ferroportin in an in vitro assay. However, it failed to elicit biologic responses in vivo, presumably due to its conversion to inactive metabolites in the bloodstream [180]. Another screen identified sulfonyl quinoxaline, a synthetic compound that likewise reacts with C326 in ferroportin and prevents its hepcidin-mediated internalization in vitro [181]; the in vivo efficacy of this drug is currently unknown. The screen also identified compounds that inhibit ferroportin internalization without interfering with the binding of hepcidin; however, these were shown to be non-specific [181].

Eli Lily and Company developed LY2928057, a humanized monoclonal ferroportin antibody that recognizes the fifth extracellular loop of ferroportin and thereby precludes the binding of hepcidin. LY2928057 maintains iron efflux from ferroportin in the presence of hepcidin, and promotes a dose-dependent increase in serum iron and hepcidin levels in cynomolgus monkeys [182]. The antibody recently completed phase 1 clinical trials and was well tolerated in hemodialyzed CKD patients. As expected, it caused an increase in serum iron and transferrin saturation; however, the effects on hemoglobinization were modest [183]. Further monoclonal antibodies against human ferroportin that recognize epitopes in the fifth extracellular loop were developed by Amgen. They were shown to partially prevent hepcidin-mediated ferroportin internalization [181], but their therapeutic potential has not been evaluated yet.

## 13. Conclusions

Within a few years after its discovery in 2001, hepcidin emerged as a therapeutic target for the treatment of iron-related “hepcidinopathies”, and possibly also for metabolic and infectious disorders. The clinical relevance for pharmacological manipulation of hepcidin pathways was established by experiments in preclinical animal models of HH, iron-loading anemias, IRIDA and AI. This led to the development of hepcidin therapeutics. Drugs that enhance or reduce hepcidin action are currently being tested in randomized controlled trials and some of them are expected to reach the clinic.

## Figures and Tables

**Figure 1 pharmaceuticals-11-00127-f001:**
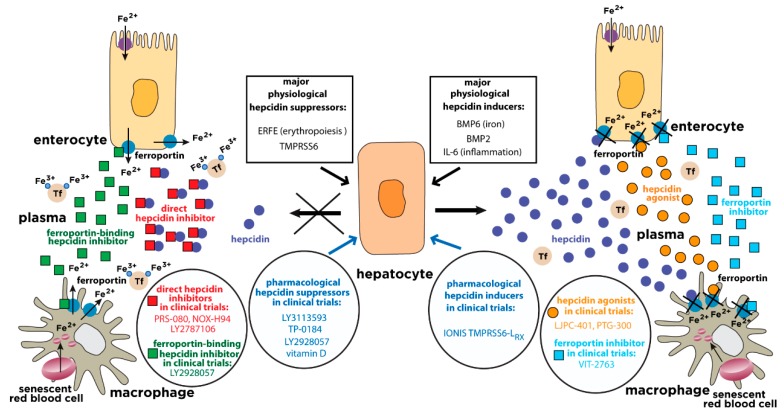
Physiological and pharmacological regulation of hepcidin activity. Hepcidin is synthesized in hepatocytes in response to iron or inflammatory stimuli. It operates by targeting the iron exporter ferroportin in tissue macrophages and duodenal enterocytes. The binding of hepcidin promotes ferroportin degradation, iron retention in target cells and hypoferremia. Suppression of hepcidin by erythropoietic stimuli allows iron efflux from cells into plasma via ferroportin. The expression of hepcidin can be manipulated pharmacologically with drugs that target hepatocytes (blue arrows). Hepcidin responses can be mimicked by hepcidin agonists (orange circles) or ferroportin inhibitors (light blue squares). Conversely, hepcidin responses can be antagonized by direct hepcidin inhibitors (red squares) or ferroportin-binding hepcidin inhibitors (green squares). The insets highlight major physiological hepcidin inducers or suppressors, as well as clinically relevant drugs that modulate the hepcidin-ferroportin axis and are currently being evaluated in randomized controlled trials for the treatment of hepcidin-related disorders (“hepcidinopathies”). Tf, transferrin.

**Figure 2 pharmaceuticals-11-00127-f002:**
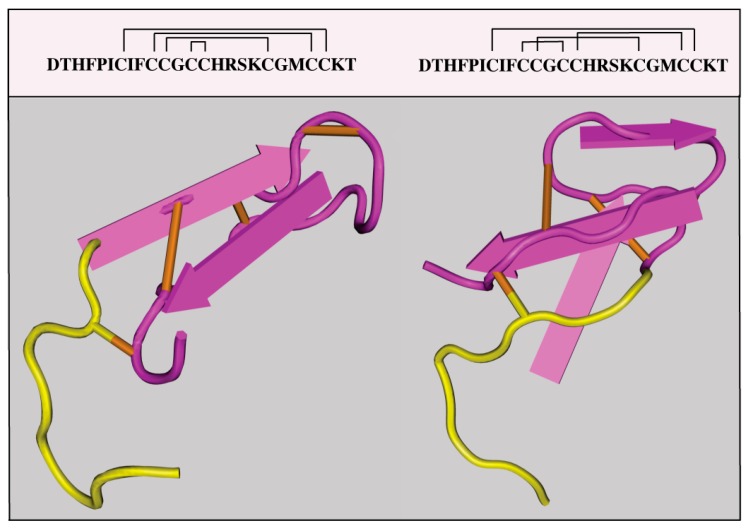
Proposed crystal structures of hepcidin. A structure with disulfide bonds between C7-C23, C10-C22, C13-C14 and C11-C19 (PDB ID: 1M4F) is shown on the left. An alternative structure with disulfide bonds between C7-C23, C10-C13, C11-C19 and C14-C22 (PDB ID: 2KEF) is shown on the right. The N-terminal amino acids which are essential for binding to ferroportin are highlighted in yellow.

**Figure 3 pharmaceuticals-11-00127-f003:**
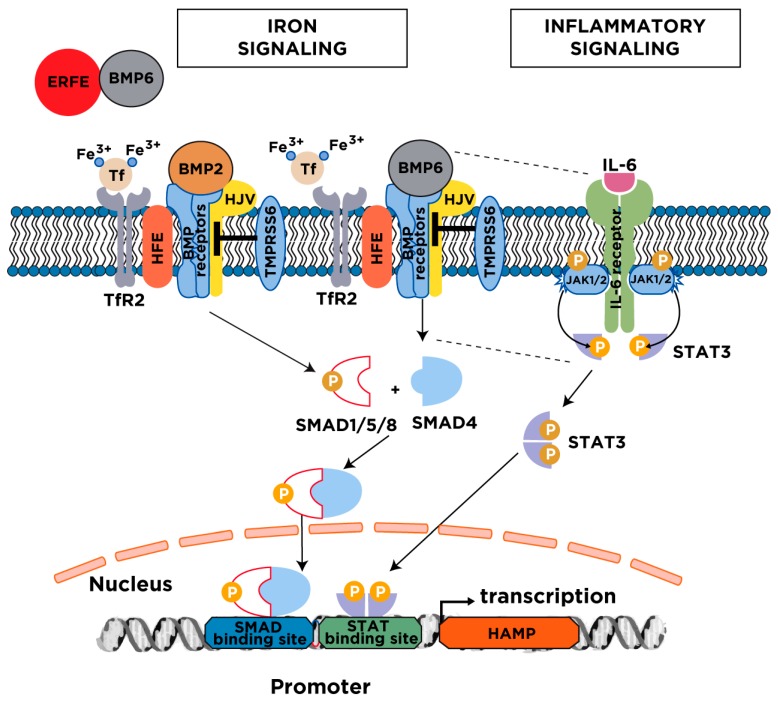
Major mechanisms for hepcidin regulation. Serum and tissue iron induce hepcidin transcription via the BMP/SMAD signaling pathway. The cascade is initiated following an increase in transferrin saturation and the secretion of BMP6 from liver sinusoidal endothelial cells; BMP2 is likewise secreted from liver sinusoidal endothelial cells but is less responsive to iron. Diferric transferrin binds to TfR2, while BMP6 and BMP2 bind to type I and II BMP receptors on hepatocytes. These events trigger phosphorylation of regulatory SMAD1/5/8, recruitment of SMAD4, and translocation of the SMAD complex to the nucleus for activating hepcidin transcription upon binding to BMP response elements in the *HAMP* promoter. Efficient iron signaling to hepcidin requires the BMP co-receptor HJV and the hemochromatosis protein HFE, and is negatively regulated by the transmembrane serine protease matriptase-2 (TMPRSS6). Under conditions of high iron demand for erythropoiesis, the erythropoietic regulator erythroferrone (ERFE) is released from bone marrow erythroblasts and suppresses hepcidin by sequestering BMP6. The inflammatory cytokine IL-6 induces hepcidin transcription via the JAK/STAT3 signaling pathway. The binding of IL-6 triggers dimerization of IL-6 receptors on hepatocytes, which leads to activation of associated JAK1/2 and subsequent phosphorylation of STAT3. Phospho-STAT3 dimerizes and translocates to the nucleus, where it activates hepcidin transcription upon binding to a STAT binding site in the *HAMP* promoter. Efficient hepcidin induction by the inflammatory pathway requires a threshold of BMP6/SMAD signaling (indicated by the dotted lines). BMP, Bone Morphogenetic Protein; SMAD, Small Mothers Against Decapentaplegic; HFE, high iron (Fe); HJV, hemojuvelin; TfR2, transferrin receptor 2; JAK, Janus kinase; STAT, Signal Transducer and Activator of Transcription.

**Table 1 pharmaceuticals-11-00127-t001:** Iron-related disorders of hepcidin dysregulation. Deficiency or excess of hepcidin has a different physiological effect on intestinal iron absorption, iron release by macrophages, serum iron, and tissue iron. HH, Hereditary Hemochromatosis; CLD, Chronic Liver Disease; IRIDA, Iron-Refractory Iron Deficiency Anemia; AI, Anemia of Inflammation.

	Disorders								
	Hepcidin Deficiency			Hepcidin Resistance or Ferroportin Deficiency		Systemic Hepcidin Overexpression			Local Hepcidin Overexpression
	HH	Iron-Loading Anemias	CLD	Ferroportin Hemochromatosis	Ferroportin Disease	IRIDA	AI	Castleman Disease	Cancer
Hepcidin	↓	↓	↓	↑	↑	↑	↑	↑	↑ (in cancer cells)
Intestinal Fe absorption	↑	↑	↑	↑	↑	↓	↓	↓	Normal
Macrophage Fe release	↑	↑	↑	↑	↓	↓	↓	↓	Normal
Serum Fe	↑	↑	↑	↑	↓	↓	↓	↓	Normal
Tissue Fe	↑	↑	↑	↑	↑	Normal	Normal	Normal	Normal

**Table 2 pharmaceuticals-11-00127-t002:** Inducers of hepcidin expression. Nrf2, nuclear factor (erythroid-derived 2)-like 2; PTGIS, prostaglandin I2 (prostacyclin) synthase; MAPK, mitogen-activated protein kinase; PRMC-1, progesterone receptor membrane component-1; PKA, protein kinase A.

	Drug	Target	Evidence	Reference
**In vitro studies**	Genistein (small molecule)	STAT3	Hepatoma cells	Zhen et al. 2013 [72]
	Ipriflavone (small molecule)	Histone deacetylase, BMP-, STAT3-dependent genes	Hepatoma cells	Gaun et al. 2014 [73]
	Vorinostat (small molecule)	Histone deacetylase, BMP-, STAT3-dependent genes	Hepatoma cells	Gaun et al. 2014 [73] Mleczko-Sanecka et al. 2017 [75]
	Diclofenac (small molecule)	Not specified; independent of PTGIS and cyclooxygenases	Hepatoma cells	Mleczko-Sanecka et al. 2017 [75]
	Icariin (small molecule)	SMAD1/5/8, STAT3	Hepatoma cells	Zhang et al. 2016 [77]
	Resveratrol, querqetin, kaemferol, naringenin, epi-galoo-catechin-3-gallate (small molecules)	Nrf2	Hepatoma cells	Bayele et al. 2015 [78]
	Sorafenib, wortmannin, rapamycin, metformin (small molecules)	Ras/RAF/MAPK and mTOR signaling	Hepatoma cells, Primary hepatocytes	Mleczko-Sanecka et al. 2014 [79]
**Preclinical studies**	BMP6	BMP receptors	Mouse model of adult HH	Corradini et al. 2010 [64]
	siRNAs	Matriptase-2	Mouse model of adult HH Mouse model of β-thalassemia	Schmidt et al. 2013 [65]
	Antisense oligonucleotides (ASOs)	Matriptase-2	Mouse model of β- thalassemia	Guo et al. 2013 [66]
	GalNac-ASOs	Matriptase-2	Mouse model of β- thalassemia, splenectomised	Schmidt et al. 2018 [70]
	Progesterone, epitiostanol, mifepristone	PRMC-1	Zebrafish	Li et al. 2016 [71]
	Ipriflavone (small molecule)	BMP-, STAT3-dependent genes	Wild type mice	Gaun et al. 2014 [73]
	Icariin (small molecule)	SMAD1/5/8, STAT3	Wild type mice	Zhang et al. 2016 [77]
	Epimedin C (small molecule)	SMAD1/5/8, STAT3	Wild type mice	Zhang et al. 2016 [77]
	Resveratrol, querqetin, kaemferol, naringenin, epi-gallo-catechin-3-gallate (small molecules)	Nrf2	Wild type rats	Bayele et al. 2015 [78]
	Adenine (small molecule)	SMAD1/5/8 and cAMP/PKA	Mouse model of adult HH	Zhang et al. 2018 [80]
**Clinical trials**	IONIS-TMPRSS6-Lrx (Antisense oligonucleotide) By: Ionis Pharmaceuticals Inc.	Matriptase-2	Healthy subjects—Phase 1 (Active)	ClinicalTrials.gov Identifier: NCT03165864

**Table 3a pharmaceuticals-11-00127-t003a:** Hepcidin agonists.

	Drug	Target	Evidence	Reference
**Preclinical studies**	Palmitoyl-ri-hep9 (minihepcidin)	Ferroportin	Mouse model of juvenile HH	Preza et al. 2011 [83]
	PR65 (minihepcidin)	Ferroportin	Wild type mice Mouse model of juvenile HH	Ramos et al. 2012 [84]
	PR73 (minihepcidin)	Ferroportin	Mouse model of adult HH Mouse model of adult HH, bacteria-infected	Arezes et al. 2015 [88] Lunova et al. 2017 [86] Stefanova et al. 2017 [89] Michels et al. 2017 [90] Stefanova et al. 2018 [91]
	M004, M009 (minihepcidins)	Ferroportin	Mouse model of β-thalassemia Mouse model of polycythemia vera	Casu et al. 2016 [92]
**Clinical trials**	LJPC-401 (hepcidin formulation) By: La Jolla Pharmaceutical Company	Ferroportin	HH, β-thalassemic patients-Phase 1 (Completed) Phase 2 (Recruiting)	Phase 1: Lal et al. 2018 [93] (congress presentation) Phase 2: ClinicalTrials.gov Identifiers NCT03395704, NCT03381833
	PTG-300 (hepcidin formulation) By: Protagonist Therapeutics Inc.	Ferroportin	Healthy subjects—Phase 1 (Completed) Phase 2 expected to start end of 2018	Bourne et al. 2018 [94] Nicholls et al. 2018 [95] (congress presentations)

**Table 3b pharmaceuticals-11-00127-t003b:** Ferroportin inhibitors.

	Drug	Target	Evidence	Reference
**Clinical trials**	VIT-2763 (small molecule) By: Vifor Pharma	Ferroportin	Phase 1 planned in 2018	http://www.viforpharma.com/en/media/press-releases/201802/2167159

**Table 4 pharmaceuticals-11-00127-t004:** Inhibitors of hepcidin expression. ALK, activin receptor-like kinase; IL-6, interleukin 6; EGFR, epidermal growth factor receptor; CKD, chronic kidney disease.

	Drug	Target	Evidence	Reference
**In vitro studies**	Genistein (small molecule)	BMP6	Macrophages	Abreu et al. 2018 [97]
	Erythroferrone	BMP6	Hepatoma cells	Arezes et al. 2018 [24]
	sHJV.Fc (antibody-like fused protein)	BMP6	Hepatoma cells, Kidney cells	Babitt et al. 2007 [101] Andriopoulos et al. 2009 [102]
	Dorsomorphin (small molecule)	Type I BMP receptors (ALK2/3/6)	Hepatoma cells	Yu et al. 2008 [106]
	LDN-193189 (dorsomorphin derivative)	Type I BMP receptors (mainly ALK2)	Primary hepatocytes	Theurl et al. 2011 [15]
	LDN-212854 (dorsomorphin derivative)	Type I BMP receptors (mainly ALK2)	Hepatoma cells	Mohedas et al. 2013 [111]
	Spironolactone (aldosterone antagonist used to treat hypertension)	BMP/SMAD signaling?	Hepatoma cells Primary hepatocytes	Mleczko-Sanecka et al. 2017 [75]
	Imatinib (tyrosine kinase inhibitor used in cancer therapy)	BMP/SMAD signaling?	Hepatoma cells Primary hepatocytes	Mleczko-Sanecka et al. 2017 [75]
	AG490, PpYLKTK, curcumin (small molecules)	STAT3	Differentiated hepatocytes	Fatih et al. 2010 [131]
	Aspirin (cyclooxygenase inhibitor for pain treatment)	JAK2, STAT3	Microglia cells Pheochromocytoma cells	Li et al. 2016 [134] Huang et al. 2018 [135]
	*Angelica sinensis* polysaccharide (small molecule)	SMAD4, STAT3/5	Hepatoma cells	Wang et al. 2017 [139]
	GDP	STAT3	Hepatoma cells Colorectal adenocarcinoma cells	Angmo et al. 2017 [145]
	17β-Estradiol	Estrogen responsive promoter	Hepatoma cells	Yang et al. 2012 [156]
	Calcitriol	Vitamin D receptor	Hepatoma cells Leukemia cells	Bacchetta et al. 2014 [161] Zughaier et al. 2014 [162]
**Preclinical studies**	Heparin	BMP6	Wild type mice	Poli et al. 2011 [96]
	Glycol-split heparin	BMP6	Mouse model of AI BMP6 knockout mice	Poli et al. 2014 [98]
	Oversulfated heparin	BMP6	Mouse model of AI	Poli et al. 2014 [99]
	sHJV.Fc (antibody-like fused protein)	BMP6	Mouse model of human SPTB Rat model of AI	Babitt et al. 2007 [101] Andriopoulos et al. 2009 [102] Theurl et al. 2011 [15]
	ABT-207 (monoclonal Ab)	HJV	Wild type rats Cynomolgus monkeys	Boser et al. 2015 [103]
	H5F9-AM8 (monoclonal Ab)	HJV	Mouse models of IRIDA and IA Rat model of AI, wild type rats Cynomolgus monkeys	Boser et al. 2015 [103] Kovac et al. 2016 [104]
	Dorsomorphin (small molecule)	Type I BMP receptors (ALK2/3/6)	Zebrafish embryos	Yu et al. 2008 [106]
	LDN-193189 (dorsomorphin derivative)	Type I BMP receptors (ALK2/3/6)	Rat model of AI Mouse model of AI	Theurl et al. 2011 [15] Theurl et al. 2014 [61] Mayeur et al. 2015 [107]
	Myricetin	SMAD 1/5/8	Wild type mice	Mu et al. 2016 [112]
	DS79182026 (small molecule)	ALK2	Mouse model of AI	Fukuda et al. 2017 [115] Sasaki et al. 2018 [116]
	TP-0184 (small molecule)	ALK2	Mouse model of AI	Peterson et al. 2015 [118] Peterson et al. 2016 [117]
	Momelotinib (JAK1/2 inhibitor for myelofibrosis treatment)	ALK2	Rat model of AI	Asshoff et al. 2017 [121]
	Spironolactone (aldosterone antagonist used to treat hypertension)	BMP/SMAD signaling?	Wild type mice	Mleczko-Sanecka et al. 2017 [75]
	Imatinib (tyrosine kinase inhibitor used in cancer therapy)	BMP/SMAD signaling?	Wild type mice	Mleczko-Sanecka et al. 2017 [75]
	Tocilizumab (monoclonal Ab for rheumatoid arthritis treatment)	IL-6	Cynomolgus monkey model of AI Mouse model of cancer anemia	Hashizume et al. 2010 [129] Noguchi-Sasaki et al. 2016 [130]
	MR16-1 (monoclonal Ab)	IL-6	Mouse model of cancer anemia	Noguchi-Sasaki et al. 2016 [130]
	AG490 (small molecule)	STAT3	Wild type mice	Zhang et al. 2011 [132]
	Maresin 1 (ω-3 fatty acid derivative)	STAT3	Mouse model of AI	Marcon et al. 2013 [136] Wang et al. 2016 [137]
	*Angelica sinensis* polysaccharide (small molecule)	SMAD4, STAT3/5	Rat model of IDA Rat model of AI	Liu et al. 2012 [138] Wang et al. 2017 [139] Wang et al. 2018 [140]
	H_2_S (gasotransmitter)	JAK2/STAT3	Mouse model of AI	Xin et al. 2016 [142] Wang et al. 2017 [143]
	GDP	STAT3	Mouse model of AI	Angmo et al. 2017 [145]
	Testosterone	SMAD1/4 or EGFR signaling	Liver-specific hepcidin-overexpressing mice Bmp6-/- mice Mouse model of AI	Guo et al. 2013 [147] Latour et al. 2014 [146] Guo et al. 2016 [149]
**Clinical trials**	LY3113593 (monoclonal Ab) By: Eli Lilly and Company	BMP6	Healthy subjects, CKD patients—Phase 1 (Completed)	ClinicalTrials.gov Identifiers: NCT02144285, NCT02604160
	sHJV.Fc (antibody-like fused protein) By: FerruMax Pharmaceuticals	BMP6	CKD patients—Phase 1 (Discontinued)	ClinicalTrials.gov Identifiers: NCT01873534, NCT02228655
	TP-0184 (small molecule) By: Tolero Pharmaceuticals Inc.	ALK2	Advanced solid tumor patients—Phase 1 (Active)	ClinicalTrials.gov Identifier: NCT03429218
	Momelotinib (JAK1/2 inhibitor used to treat myelofibrosis)	ALK2	Myelofibrosis patients—Phase 1/2 (Completed)	Pardanani et al. 2013 [119]
	Tocilizumab (monoclonal Ab for rheumatoid arthritis treatment)	IL-6	Rheumatoid arthritis patients Castleman disease patients	Song et al. 2010 [127] Song et al. 2013 [124] Isaacs et al. 2013 [125] Suzuki et al. 2017 [126]
	Siltuximab (monoclonal Ab for neoplastic disease treatment)	IL-6	Castleman disease patients	Casper et al. 2015 [128]
	Curcumin (small molecule)	STAT3	Healthy subjects	Laine et al. 2017 [133]
	Testosterone	SMAD1/4 or EGFR signaling	Type 2 diabetes patients with hypogonadotropic hypogonadism	Dhindsa et al. 2016 [153]
	17β-Estradiol	Estrogen responsive promoter	Patients with growth hormone deficiency/hyperthyroidism/ hyperprolactinemia Premenopausal women	Lehtihet et al. 2016 [157] Bajbouj et al. 2018 [158]
	Vitamin D_2_	Vitamin D receptor	Healthy subjects	Bacchetta et al. 2014 [161]
	Vitamin D_3_	Vitamin D receptor	CKD patients Healthy subjects Critically ill patients	Zughaier et al. 2014 [162] Smith et al. 2017 [163] Smith et al. 2018 [164]

**Table 5 pharmaceuticals-11-00127-t005:** Hepcidin antagonists.

	Drug	Target	Evidence	Reference
**In vitro studies**	Fursultiamine (small molecule)	Ferroportin	Kidney cells	Fung et al. 2013 [180]
	Quinoxaline (small molecule)	Ferroportin	Kidney cells, breast cells, leukemia cells	Ross et al. 2017 [181]
**Preclinical studies**	LY2928057 (monoclonal Ab)	Ferroportin	Cynomolgus monkeys	Witcher et al. 2013 [182]
**Clinical trials**	LY2787106 (monoclonal Ab) By: Eli Lily and Company	Hepcidin	Patients with cancer-associated anemia—Phase 1 (Completed)	Vadhan-Raj et al. 2017 [170]
	PRS-080 (Pegylated anticalin) By: Pieris Pharmaceuticals GmbH	Hepcidin	Anemic CKD patients—Phase 1b/2a (Recruiting)	ClinicalTrials.gov Identifiers: NCT02754167, NCT03325621
	NOX-H94 (Pegylated spiegelmer)	Hepcidin	Healthy subjects—Phase 1 (Completed) Endotoxemia-induced in volunteers—Phase 1 (Completed) Patients with cancer-associated anemia—Phase 2a (Completed) ESA-hyporesponsive anemia in CKD patients—Phase 2b (Completed)	Boyce et al. 2016 [176] Van Eijk et al. 2014 [177] Macdougall et al. 2015 [178] Georgiev et al. 2014 [179]
	LY2928057 (monoclonal Ab) By: Eli Lily and Company	Ferroportin	Healthy subjects and hemodialyzed patients—Phase 1 (Completed)	Barrington et al. 2016 [183]

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
