# Peer review of "Hepcidin Therapeutics"

_pharmaceuticals, 2018, doi:10.3390/ph11040127_

Round 1

Reviewer 1 Report

The review presents an overview on systemic iron metabolism focusing on the chemical and molecular features of iron hormone hepcidin and on a detailed summary of the different therapeutic approaches aimed to modulate hepc amount in common iron disorders.

I believed that it gives comprehensive information about a quite recent field of scientific investigation like hepcidin based therapy with appropriate and updated references.

My minor remarks would be:

Abstract: pg 1 line 12 is a pathogenic cofactor

Table 1. small and difficult to read, I would suggest to use abbreviations explained below the table ( es HH for hereditary  hemochromatosis) and to increase the arrows size.

Tables 2-4. Could be possible to add the ref number after the first author surname in the reference column? I would make the full reference easy to find in the specific section.

Author Response

We thank the reviewer for his/her positive comments. We modified the abstract, Table 1 and Tables 2-4 as suggested.

Reviewer 2 Report

The authors present an excellent overview of iron metabolism focusing on hepcidin as targetable iron regulatory hormone. They manage to provide a comprehensive review of the current literature with fantastic illustrations and sophisticated tables. Therefore the review is highly suitable for publication in ‚Pharmaceuticals’.

I have only few minor comments to make:

Figure 1: The authors could consider indicating the effect of a ferroportin-binding hepcidin inhibitor on enterocytes and macrophages with an arrow in bolt or two arrows.

Are all effects of EPO mediated by ERFE or does EPO still qualify as major physiological hepcidin suppressor?

Table 1 (and section 6). In malignancy, I see two possible scenarios: In limited disease, cancer cells may locally produce hepcidin leading to iron-redistribution in the tumor microenvironment (very last column from the right in table 1). In advanced disease stages, the immune response against tumor cells including metastatic ones may result in systemic overproduction of hepcidin and AI (third last column from the right in table 1). This may result in anemia based on reduced iron absorption and recycling. This could be highlighted further in the main text e.g. by describing anemia of cancer as a variant of anemia of inflammation or by discussing malignancy as a spectrum from localized to advanced stages.

The term ‚hepcidinopathy’ is novel and deserves clear definition when first mentioned in the main text.

Line 12: ‚…is a pathogenic cofactor…’

Line 116. ‚erythropoiesis‘

Lines 141, 240, 242. The type of diabetes mellitus associated with HH is better classified as a ‚specific type of diabetes due to other causes’.

Lines 290 and 573. ‚treatment‘/‚treated‘ or ‚inoculation‘/‚inoculated‘

Line 305. Please discuss the side effect in the context of the route of application.

Line 511. ‚hypertension and congestive heart failure’

Line 601. ‚critically ill’

Author Response

We thank the reviewer for his/her positive comments and insightful suggestions. We made all recommended modifications and updated Fig. 1 for clarity. In addition, we added in the text some information about the ferroportin inhibitor from Vifor Pharma. Even though not much is known about this molecule, it represents a new class of drugs; thus, we decided to also include it in revised Fig. 1.